# 2’-Fucosyllactose Inhibits Coxsackievirus Class A Type 9 Infection by Blocking Virus Attachment and Internalisation

**DOI:** 10.3390/ijms232213727

**Published:** 2022-11-08

**Authors:** Fuxing Lou, Ruolan Hu, Yangzhen Chen, Mengzhe Li, Xiaoping An, Lihua Song, Yigang Tong, Huahao Fan

**Affiliations:** College of Life Science and Technology, Beijing University of Chemical Technology, Beijing 100029, China

**Keywords:** 2’-fucosyllactose, CV-A9, milk, enterovirus

## Abstract

Coxsackieviruses, a genus of enteroviruses in the small RNA virus family, cause fatal infectious diseases in humans. Thus far, there are no approved drugs to prevent these diseases. Human milk contains various biologically active components against pathogens. Currently, the potential activity of breast milk components against the coxsackievirus remains unclear. In our study, the inhibitory effect of 16 major human milk components was tested on coxsackievirus class A type 9 isolate (CV-A9), BUCT01; 2’-Fucosyllactose (2’-FL) was identified to be effective. Time-of-addition, attachment internalisation assays, and the addition of 2’-FL at different time points were applied to investigate its specific role in the viral life cycle. Molecular docking was used to predict 2’-FL’s specific cellular targets. The initial screening revealed a significant inhibitory effect (99.97%) against CV-A9 with 10 mg/mL 2’-FL, with no cytotoxicity observed. Compared with the control group, 2’-FL blocked virus entry (85%) as well as inhibited viral attachment (48.4%) and internalisation (51.3%), minimising its infection in rhabdomyosarcoma (RD) cells. The cell pre-incubation with 2’-FL exhibited significant inhibition (73.2–99.9%). Extended incubation between cells with 2’-FL reduced CV-A9 infection (93.9%), suggesting that 2’-FL predominantly targets cells to block infection. Molecular docking results revealed that 2’-FL interacted with the attachment receptor α_v_β_6_ and the internalisation receptor FCGRT and β_2_M with an affinity of −2.14, −1.87, and −5.43 kcal/mol, respectively. This study lays the foundation for using 2’-FL as a food additive against CV-A9 infections.

## 1. Introduction

The gastrointestinal immune system of newborns is weak, increasing their susceptibility to pathogen attacks. Breast milk contains rich active ingredients that offer energy and nutrition, regulating immune-related homeostasis and providing a protective mechanism for inducing tolerance to antigens in early human development [1]. Additionally, the antiviral properties of breast milk have been confirmed. After incubating with breast milk for 30 min at room temperature, the virus titre of the five severe acute respiratory syndrome coronavirus 2 (SARS-CoV-2) strains decreased by 40.9–92.8% [2]. A 100–1000-fold reduction in live virus titre was detected after pre-incubating hepatitis C virus (HCV) with breast milk for 1 h [3]. This was facilitated by the presence of various antiviral substances, including lactoferrin (LF) [4], linoleic acid [5], IgA [6], whey protein [7], and human milk oligosaccharides (HMOs) [8].

LF has been proven to protect against numerous pathogens, including papillomavirus [9], human immunodeficiency virus [10], rotavirus [11], zika virus [12], and enterovirus [13]. Linoleic acid and whey protein can inhibit viral infection dose-dependently by interfering with the interaction between SARS-CoV-2 and angiotensin converting enzyme 2 (ACE2) [7,14]. HMOs, a class of free oligosaccharides unique to human milk, can act as receptor decoys or compete with viruses for cellular receptors. Their ability to inhibit rotavirus [8], norovirus [15], influenza virus [16], and human immunodeficiency virus [17] has been reported.

Human enterovirus B (HEV-B) is the most abundant enterovirus group, including enteric cytopathic human orphan virus (Echo), coxsackievirus group B (CV-B), and coxsackievirus class A type 9 isolate (CV-A9), and is one of the major viruses ravaging the paediatric population [18]. HEV-B infection is associated with viral encephalitis, aseptic meningitis, blistering viral stomatitis, and hand, foot, and mouth disease in children, particularly neonates, leading to heart failure, liver damage, and even death [19]. Although the genus enterovirus includes many important human pathogens, no approved drugs are currently available to prevent their infections. In addition, vaccines for enteroviruses are not widely available. The vast variety of enterovirus species with multiple variants makes vaccine designs challenging. The only vaccines available for clinical use are poliovirus (PV) and enterovirus group A type 71 (EV-A71). Considering the mutation-prone nature of small RNA viruses, it is not easy to design vaccines for all enteroviruses, highlighting the importance of drug research. Considering that breast milk contains several antiviral components, in the present study we tested the effects of various bioactive factors from breast milk on the infectivity of CV-A9 strain BUCT01.

## 2. Results

### 2.1. Five Components with Significant Anti-CV-A9 Effect Were Screened from Breast Milk

Four proteins (whey protein concentrate (WPC), milk fat globular membrane (MFGM), LF, and OPN), eight oligosaccharides (2’-FL, 3’-Fucosyllactose (3’-FL), 3’-Sialyllactose (3’-SL), 6-sialyllactose (6’-SL), lacto-N-tetraose (LNT), lacto-N-neotetraose (LNNT), fructooligosaccharide (FOS), and galactose oligosaccharides (GOS)), and four vitamins (vitamin B1, B2, D2, and D3) in breast milk were tested for their inhibitory effects on CV-A9 (all components were dissolved in DMEM) (Figure 1). The final concentrations of all components used for primary screening are indicated in the methods section. No components showed significant cytotoxicity at the primary screening concentrations (Appendix A).

The top five most effective agents (LF, 2’-FL, MFGM, WPC, and OPN) are indicated by blue bars in Figure 1. LF inhibits infection by directly attaching viral particles or binding to cellular receptors. It may also restrain viral replication in host cells by activating immune cells or cytokines [4,20,21,22]. Cellular assays in our study revealed a significant anti-CV-A9 effect of LF (EC_50_ = 0.01092 mg/mL, CC_50_ > 5 mg/mL) (Appendix A). The time-of-addition assay showed that 0.04 mg/mL LF can effectively inhibited viral entry and replication, and can be used as a positive control in subsequent experiments to characterise other compounds’ antiviral properties (Appendix A).

Another component, 2’-FL, with a significant anti-CV-A9 effect screened in our study, showed almost complete viral RNA inhibition in RD cells at final concentrations of 10 mg/mL (99.97%) and 5 mg/mL (99.91%) (Figure 2A). In addition, we tested the sensitivity of the virus to 2’-FL in two cell lines, RD (EC_50_ = 0.8956 mg/mL, CC_50_ > 20 mg/mL) and Huh7 (EC_50_ = 6.955 mg/mL, CC_50_ > 20 mg/mL), 2’-FL’s anti-CV-A9 property was non-cell-dependent (Figure 2A,B).

Since MFGM and WPC are mixtures, and WPC’s and OPN’s inhibition efficiency on CV-A9 infection is <90%, further analysis was not performed in this study. In summary, we analysed the specific antiviral mechanisms of 2’-FL on CV-A9 infection.

### 2.2. Oligosaccharide 2’-FL Targets the Host Cell Instead of the Virus Particle to Block CV-A9 Infection

The reduced CV-A9 infectivity can be mediated by targeting cells or viruses. To determine whether 2’-FL exerts a direct effect on the CV-A9 virus, the virus was pre-incubated with 2’-FL at a final concentration of 10 mg/mL at 4 °C for 2 h. In the control group, the virus was treated with DMEM only; followed by a gradient dilution of the treated virus solution to determine the virus titre using plaque assay. The results showed that the virus titre was not reduced by 2’-FL treatment, suggesting 2’-FL does not exert its antiviral effect by directly disrupting the virus’ physical structure (Figure 2C).

Subsequently, we pre-incubated cells with 2’-FL at two final concentrations (20 mg/mL and 10 mg/mL) for 4 h, followed by CV-A9 (MOI = 0.001) infection in the presence of 2′-FL for 48 h. Compared with the control group, a significant reduction in virus load (73.2–99.99%) was observed in both 2’-FL treatment groups (Figure 3A). However, after pre-treating the virus with the same 2’-FL concentration at 4 °C for 4h, the intracellular CV-A9 load did not reduce (Figure 3B). Furthermore, similar results were observed in Huh7 cells, with considerable viral load reduction after pre-treating cells with 2’-FL (68.3–94.2%) (Figure 3A). In the virus pre-treatment group, intracellular viral load was similar to the control group (Figure 3B). In addition, extended Caco-2 cell incubation with HMOs alters glycan chain expression on the cell surface, affecting enteropathogenic E. coli attachment [23]. RD cells were treated with 2’-FL (20 mg/mL) for 24 h. Then, the supernatant was removed and replaced with CV-A9 solution (MOI = 0.001) for 24 h in the absence of 2’-FL to exclude its antiviral effect during reinfection. The results suggested that extended treatment reduced the intracellular viral load (93.9%) (Figure 3C). In conclusion, 2’-FL worked on the host cells but not the virus to block CV-A9 infection.

### 2.3. The Effect of 2’-FL on CV-A9 Life Cycle

A time-of-addition assay was performed to clarify the specific stages of the CV-A9 life cycle in which 2’-FL plays a role. Cells were treated with 2’-FL (10 mg/ml) at Full-Time, Entry, and Post-Entry during virus infection. We noticed that in the Entry-treated group, the intracellular viral mRNA load was 85% lower than that in the control, and there was a 10-fold live viral titre suppression in the supernatant. At the Post-Entry stage, 2’-FL presented a weak inhibitory effect with a 2-fold lower intracellular virus load and similar viral titres in the supernatant compared with the control group (Figure 4A). The time course and single-step growth curve revealed a negligible effect of 2’-FL on viral genome replication compared with LF (Figure 4B,C; Appendix A). These results suggest that 2’-FL mainly functioned in the virus entry stage.

We then determined the specific role of 2’-FL in virus entry by adding 2’-FL (10 mg/mL) at different infection stage (CV-A9 MOI = 5) (Figure 4D). Infections in the absence of 2’-FL served as controls, and cells were collected at 12 h.p.i. for qRT-PCR analysis. CV-A9′s internalisation occurred at 2 h.p.i.; at 3 h.p.i., the viral genome began replicating, and intracellular double-stranded RNA levels increased [24]. Therefore, our drug-delivery procedure was divided into four timepoints: pre-infection, simultaneous virus addition (binding), virus uncoating (1 h), and post-internalisation (2 h). At the pre-infection and binding stages, the virus was significantly suppressed with 73.4% and 73.6% inhibition, respectively, whereas no inhibition was observed in the other stages (Figure 4D). Subsequently, to eliminate the effect of drug duration in cells, we standardised the duration to 1 h. These results are consistent with those of previous studies, suggesting that 2’-FL specifically affects viral binding (Figure 4E).

### 2.4. Oligosaccharide 2’-FL Inhibits CV-A9 Binding and Internalisation

Previous studies have shown that CV-A9 entry into cells involves viral binding, followed by uncoating and genomic injection [24]. RDs were first put on ice for 2 h and then at 37 °C for 1h, which allows for entry steps to proceed. Virus infection and drug treatment were performed as mentioned in methods. The results showed a significant reduction in both cell-bound (48.4%) and internalized (51.3%) viral load compared with that in the control (Figure 4F,G). These results showed that 2’-FL’s inhibitory effect on CV-A9 attachment and internalisation might be crucial for its resistance to viral infection.

α_v_β_6_ is an attachment receptor for CV-A9 without the ability to induce conformational changes required for virus uncoating [25]. A genome-wide knockout using the CRISPR Cas9 library identified FCGRT and β_2_M as uncoating receptors for many enterovirus B families, including CV-A9 [20]. In our study, 2’-FL’s molecular docking with the attachment receptor α_v_β_6_ (ITGB6) and the uncoated receptors FCGRT and β_2_M was applied to predict its potential interaction with these three proteins. The docking results are shown in Table 1; 2’-FL possesses a potential binding capacity to all three proteins, consistent with the results of attachment and internalisation analysis. The potential hydrogen bonds across the interface were predicted: W51 of FCGRT, I216 of ITGB6, and Q155, D4, and R62 of β_2_M (Figure 5A–C).

## 3. Discussion

Several ingredients derived from breast milk possess antiviral activities. LF inhibits various viruses, including enveloped viruses such as SARS-CoV-2 [26,27], human hepatitis C virus [28], human hepatitis B virus [29], HIV [30], herpes viruses [31], hantaviruses [32], and non-enveloped viruses such as human papillomavirus (HPV) [9], enterovirus 71 (EV71) [13], echovirus 6 [33], adenovirus [34], and feline virus [35]. Whey protein is a mixture of α-whey protein, immunoglobulins, serum albumin, LF, and lysozyme. Numerous antiviral effects have been reported, including resistance to influenza virus [36], human rotavirus [37], and enterovirus [38]. HMOs, the third-most abundant solid component in human milk after lactose and lipids, improve the intestinal microbiota’s composition and possess antibacterial and antiviral properties. HMOs exhibit anti-streptococcal, Staphylococcus aureus, and Acinetobacter baumannii properties enhancing the antibacterial effects of aminoglycosides, anti-folate, macrolides, lincosamides, and tetracyclines [39,40]. In the antiviral field, HMOs rely on two main mechanisms to exert their effects: first, due to their similar structure to cell surface glycans, HMOs can act as soluble decoy receptors to prevent early viral infections, including norovirus and rotavirus; second, HMOs bind with cell surface receptors to block viral attachment (Figure 6). Pre-incubating 2’-FL, 3’-SL, 6’-SL, and galactooligosaccharides with viruses significantly reduced two human rotaviruses’ infectivity (G1P and G2P) on MA104 cells, whereas pre-incubating with cells produced no significant suppression [8]. In a piglet model, HMOs shortened the rotavirus-induced diarrhoea duration [41]. In addition, by interacting with the Histo-Blood Group Antigen (HBGA) binding pocket on the norovirus capsid, 2’-FL almost eliminated GI.1 and GII.17 norovirus binding to HBGAs, exerting an antiviral effect. Given the conservation of the interaction between noroviruses and HBGAs, 2’-FL may be a broad-spectrum agent against multiple noroviruses [42]. Respiratory syncytial virus (RSV), which causes respiratory and pulmonary infections in children, contains fusion proteins and attachment glycoproteins essential for early viral attachment; 2’-FL and 3’-FL bind attachment glycoproteins to reduce RSV viral load in respiratory epithelial cells [43]. Lewis blood group antigens in HMOs inhibit HIV’s binding to dendritic cell-specific ICAM3-grabbing non-integrin (DC-SIGN), exhibiting increased interaction with 2’-FL and 3’-FL [44,45]. HMOs protect against various pathogens; however, related research on anti-coxsackievirus has not been reported. In this study, we found that 2’-FL in HMOs can effectively inhibit CV-A9 for the first time and confirmed that 2’-FL exerts an anti-CV-A9 effect in vitro by targeting host cells. Unlike previous inhibition mechanisms for norovirus or rotavirus, pre-incubation with CV-A9 did not result in reduced infectivity in RD and Huh7 cells, suggesting that 2’-FL cannot act as a decoy receptor for CV-A9. We hypothesized that there is a weak interaction between 2’-FL and virus, which is not targeting the viral active site and therefore cannot block viral binding to cell receptors. However, this weak interaction between virus and 2’-FL promotes viral arrival at the cell surface, thus increasing viral binding to the cell. A similar phenomenon was also reported for G10P rotavirus, where treatment with the same dose of mixed HMOs reduced G3P rotavirus infection but enhanced G10P rotavirus [46]. Further mechanistic studies would be needed in the future to analyse the interaction between 2’-FL and virus; 2’-FL had a negligible effect on viral replication in our experiments, acting in early viral attachment and genomic injection. The different drug treatment phases suggested that 2’-FL blocked virus binding but had no role in the internalisation phase (Figure 4D,E). However, since multiple round viral replication may occur within 12 h, weakening 2’-FL’s role in the attachment and internalisation phases, early attachment and internalisation analysis for viruses within only 2 h were applied to confirm its positive inhibitory role (Figure 4F,G).

The attachment receptor for CV-A9 is believed to be α_v_β_6_ [47]. FCGRT and β_2_M are universal uncoating receptors essential for CV-A9 infection [48]. The molecular docking results suggested that 2’-FL possessed a potential binding capacity to all three proteins, among which β_2_M had the best binding effect with a binding energy of −3.19. These results were consistent with our experimental results, as the inhibitory effect of 2’-FL in the internalisation phase was better than that of attachment (Figure 4F,G). The significant difference in 2’-FL’s anti-CV-A9 effects between the RD (EC_50_ = 0.8956 mg/mL) and Huh7 (EC_50_ = 6.955 mg/mL) cells may result from the fact that Huh7 is an α_v_β_6_-deficient cell, whereas RD has the opposite effect (Figure 2B). CV-A9 can attach to Huh7 cells in an α_v_β_6_-independent manner, whereas FCGRT and β_2_M remained necessary for infection. Therefore, 2’-FL would be less effective on Huh7 cells than RD; however, it inhibits CV-A9 infection by interacting with FCGRT and β_2_M to block viral internalisation.

In summary, we first identified that 2’-FL, with no cytotoxicity, is highly effective in inhibiting clinical coxsackievirus isolates, providing conclusive evidence for its use as a food additive against coxsackievirus infection and highlighting the potential importance of HMO addition to infant formula.

## 4. Materials and Methods

### 4.1. Cell Culture and Virus Infection

Human rhabdomyosarcoma (RD) cells obtained from the American Type Culture Collection (ATCC, Manassas, VA, USA) were maintained in Dulbecco’s modified Eagle’s medium (DMEM; Gibco, Carlsbad, CA, USA) containing 10% foetal bovine serum (FBS; PAN, Aidenbach, Germany) and 1% antibiotic-antimycotic solution (Gibco, Carlsbad, CA, USA) at 37 ℃ with 5% CO_2_.

The CV-A9 strain BUCT01 (GenBank accession No. MW192795) was isolated from the faeces of a patient with hand-foot-and-mouth disease (HFMD) in China and identified using high-throughput sequencing as a coxsackievirus A9 strain.

RD cells were infected with CV-A9 at a 0.001 multiplicity of infection (MOI). At 48 h post-infection (h.p.i.), the culture supernatant was collected after centrifuging at 1000× *g* for 3 min. The viral CV-A9 titres were determined using a plaque assay.

### 4.2. Preliminary Drug Screening In Vitro

RD cells were seeded in 96-well plates. At 80–90% cell density, CV-A9 (MOI = 0.001) and various drugs (2’-FL [10 mg/mL], 3’-Fucosyllactose [10 mg/mL], lacto-N-Tetraose [10 mg/mL], lacto-N-Neotetraose [10 mg/mL], 3’-Sialyllactose [10 mg/mL], 6’-Sialyllactose [10 mg/mL], whey protein concentrate [5 mg/mL], milk fat globule membrane [5 mg/mL], LF [5 mg/mL], OPN [5 mg/mL], Vitamin B2 [100 ug/mL], Vitamin B1 [1 mg/mL], Vitamin D2 [1 mg/mL], and Vitamin D3 [1 mg/mL]) were added. At 2 h.p.i., the virus–drug mixture was removed, and fresh culture medium containing the same drug dosages as before was added. At 48 h.p.i., the cells were collected for viral yield quantification using quantitative real-time PCR (qRT-PCR).

The relative viral mRNA expression and inhibition rate were calculated based on the Ct value, denoted by B and I, respectively. The formula is as follows:

A = expression level of CV-A9 mRNA/GAPDH mRNA = 2^(CtGAPDH − CtCV-A9);^

B_ingredient_ (%) = (A_ingredient_/A_control_) × 100%;

I_ingredient_ (%) = 1 − (B_ingredient_ − B _control_) × 100%;

Active ingredients with inhibition rates >90% are potential antiviral ingredients. Further verification of antiviral activity and preliminary mechanistic studies of these potential antiviral ingredients is conducted in the present study.

### 4.3. Antiviral and Cytotoxicity Assays

Cells were co-incubated with CV-A9 (MOI = 0.001) and different 2’-FL doses at final concentrations of 40, 20, 10, 5, 2.5, 1.25, 0.625, 0.3125, 0.156, and 0.078 mg/mL. At 2 h.p.i., the virus–drugs mixture was removed, and a fresh culture medium containing the same drug dosages as before was added to each well. At 48 h.p.i., all cells were harvested for RNA extraction and 50% effective concentration (EC_50_) assay.

Cells were incubated with different 2’-FL doses at final concentrations of 40, 20, 10, 5, 2.5, 1.25, 0.625, 0.3125, 0.156, and 0.078 mg/mL for 48 h. The 50% cytotoxic concentration (CC_50_) of compounds to cells and cell viability % was measured using Cell-Titer-Blue (Promega, Cat No. G8088, Madison, WI, USA) with a Synergy H1 microplate reader (BioTek, Winoosk, VT, USA).

Cells were co-incubated with 2’-FL for 4h at final concentrations of 20 mg/mL, 10 mg/mL respectively at 37 °C, followed by infection with CV-A9 (MOI = 0.001) in the presence of the same dosages of 2’-FL as before. At 2 h.p.i., the virus–drugs mixture was removed, and a fresh culture medium containing the same drug dosages as before was added to each well. At 48 h.p.i., all cells were harvested for RNA extraction.

After pre-diluting the virus solution to MOI = 0.002, the virus solution was incubated with 2’-FL (volume ratio 1:1) for 4 h at 4 °C to ensure the final drug concentration of 20 mg/mL and 10 mg/mL, and the final viral MOI = 0.001. The virus–drugs mixture was then added to the cells. At 2 h.p.i., the virus–drugs mixture was removed, and a fresh culture medium containing the same drug dosages as before was added to each well. At 48 h.p.i., all cells were harvested for RNA extraction. Cells without drug treatment and only with virus treatment were the control group.

### 4.4. Time-of-Addition Assay

RD cells were treated with 2’-FL (10 mg/mL) at different CV-A9 infection stages. The Full time, Entry, and Post-Entry treatment experiments were performed as previously reported [49]. For Full time and Entry treatments, 2’-FL was added to the cell culture medium for a 1 h pre-treatment before virus attachment. After 24 h.p.i. viral RNA was collected from the cells for qRT-PCR analysis, and the viral load in the supernatant was measured by plaque assay.

### 4.5. Viral RNA Extraction and qRT-PCR

RD cells were harvested for RNA extraction using the AxyPrepTM Body Fluid Viral DNA/RNA Miniprep Kit (Cat No. AP-MN-BF-VNA-250, Hangzhou, Zhejiang, China) and the AxyPrepTM Multisource Total RNA Miniprep Kit (Axygene, Cat No. AP-MN-MS-RNA-250G) according to the manufacturer’s instructions. Reverse transcription was performed using a Hifair II 1st Strand cDNA Synthesis Kit with a gDNA digester (Yeasen Biotech, Cat No.11121ES60, Shanghai, China). A qRT-PCR was performed using the QuantStudio 1 Real-Time PCR detection system (Applied Biosystems, Foster City, CA, USA) with Hieff qPCR SYBR Green Master Mix (Yeasen Biotech, Cat:11202ES08, Shanghai, China). The primer sequences used for qRT-PCR are listed in Table 2.

### 4.6. Virus Attachment and Internalisation Assay

RD cells were seeded in 24-well plates and treated with CV-A9 or 2’-FL at 90–100% confluency. To determine the effect of 2’-FL on viral attachment, cells were co-incubated with CV-A9 (MOI = 5) and 2’-FL (10 mg/mL) at 4 °C for 2 h. The cells were then washed twice with phosphate-buffered saline (PBS) to remove unbound viruses. To determine CV-A9 internalisation, the cells were co-incubated with CV-A9 (MOI = 5) first on ice then washed with PBS and replaced with 2’-FL (10 mg/mL) and cultured at 37 °C for 1 h. The surface-bounded virus was removed by proteinase K digestion for 15 min at 4 °C. Then all cells were harvested for RNA extraction, the viral load was quantified using qRT-PCR and normalised to GAPDH levels.

### 4.7. Plaque Assay

Cells were seeded in 6-well plates. At 80–90% confluency, cells were infected with serially 10-fold diluted CV-A9, ranging from 10^−1^ to 10^−6^. At 2 h.p.i., the supernatant was discarded, and the unbound virus was washed off with PBS. Furthermore, a 1% agarose (2 mL per well) overlay was applied to each well to prevent cross-contamination; the wells were then incubated at 37 °C for 72 h or 96 h. The cells were fixed with 4% paraformaldehyde for 1 h at 25 ± 2 °C. The top semi-solid agarose medium was removed, and the cells were stained with crystal violet for 10 min and gently rinsed with water. The number of plaques was counted, and the virus titre was calculated.

### 4.8. Time Course and Single-Step Growth Curve Experiment

Cells were co-incubated with CV-A9 (MOI = 0.001). At 2 h.p.i., the supernatant was removed, and cells were washed twice with PBS to remove unbound viruses. Then a fresh culture medium containing 2’-FL (10 mg/mL) or LF (0.156 mg/mL) was added to each well. Cells were harvested at 4, 6, 10, 12, 24, 36, and 48 h.p.i. for qRT-PCR and time course curves assay. The supernatant harvested at 4, 6, 10, 12, 24, 36, and 48 h.p.i. contains live progeny virus, enabling analysis of single-step growth curves by plaque assay.

### 4.9. Molecular Docking Analysis

Along with several proteins (FCGRT, β_2_M, and ITGB6) 2’-FL was docked; docking was analysed using AutoDock 4.2.6 and Pymol. The FCGRT (PDB ID:1EXU), β_2_M (PDB ID:1A1M), and ITGB6 (PDB ID:4UM8) structures were obtained from PDB (https://www.rcsb.org/), in pdb format. The 2-dimensional (2D) 2’-FL structure (CID_16219342) was obtained from PubChem (https://pubchem.ncbi.nlm.nih.gov/) in sdf format.

### 4.10. Statistical Analysis

Data were analysed using GraphPad Prism 8 software (GraphPad Software Inc, San Diego, CA, USA). Comparisons between the two groups were performed using Student’s *t*-tests. A *p*-value of <0.05 was considered statistically significant. The symbols are defined as follows: ns: no significant difference; * *p* < 0.05; ** *p* < 0.01; *** *p* < 0.001; and **** *p* < 0.0001.

## Figures and Tables

**Figure 1 ijms-23-13727-f001:**
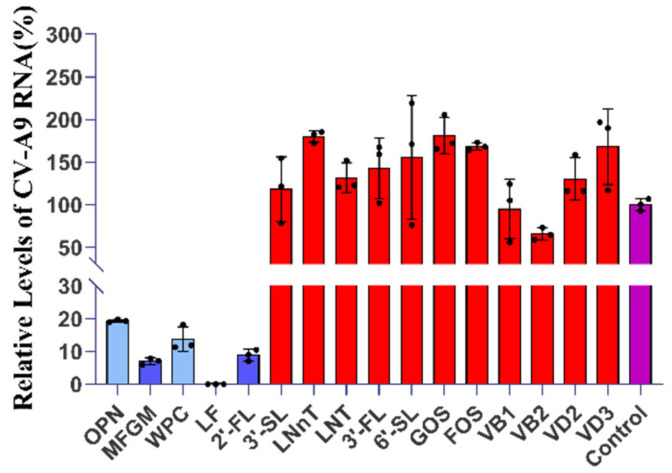
Hits of anti-CV-A9 drug screening from 16 breast milk ingredients. A total of 5 hits (MFGM, WPC, 2’-FL, LF, and OPN) were found with significant inhibition against CV-A9 indicated by the blue bars; OPN and WPC with an inhibition rate of <90% were indicated in light blue. Ineffective drugs are represented by red bars and control by purple. The Y-axis represent the relative viral load. The relative viral mRNA expression and inhibition rate were calculated based on the Ct value, denoted by B and I, respectively. The formula is as follows: A = expression level of CV-A9 mRNA/GAPDH mRNA = 2^(CtGAPDH − CtCV-A9)^; B_ingredient_ (%) = (A_ingredient_/A_control_) × 100%; I_ingredient_ (%) = 1 − (B_ingredient_ − B_control_) × 100%. Data represent the mean of the three replicate results. Error bars represent ±1 SD, and the experiments were repeated at least twice.

**Figure 2 ijms-23-13727-f002:**
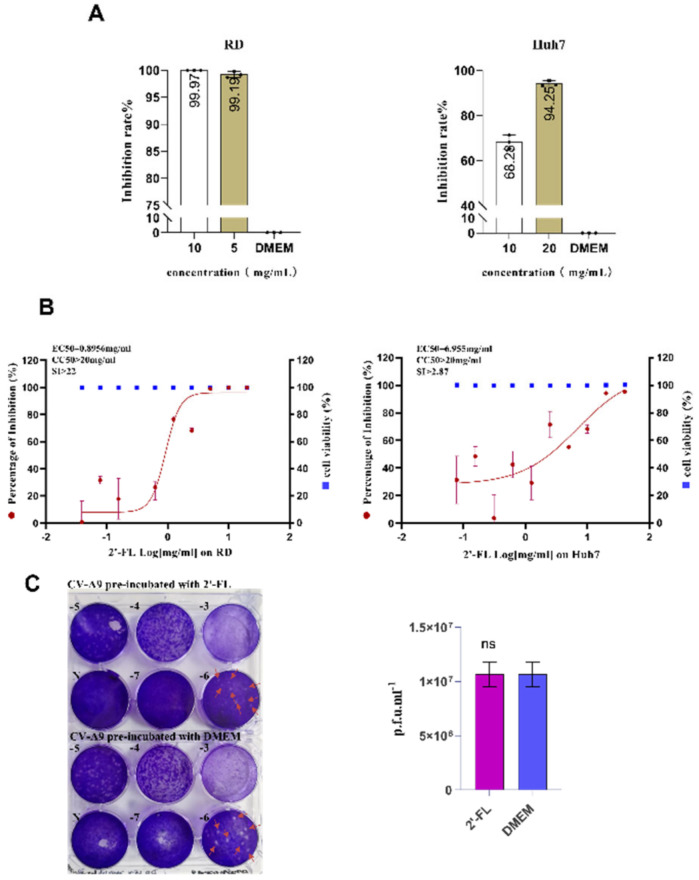
The anti-CV-A9 activity of 2’-Fucosyllactose (2’-FL). (**A**) 2’-FL’s inhibition rate against CV-A9 on RD and Huh7. Cells were infected with CV-A9 at MOI = 0.001 and treated with 2’-FL at a final concentration of 20, 10, and 5 mg/mL for 48 h. The intracellular viral load was quantified using qRT-PCR and normalised by GAPDH level. (**B**) The anti-CV-A9 EC_50_ and CC_50_ of 2’-FL on RD and Huh7. Cells were infected with CV-A9 at MOI = 0.001 and treated with 2’-FL at various concentrations (from 40 mg/mL to 0.078 mg/mL) to determine viral inhibition and cell viability. Concentration ≥5 mg/mL inhibited CV-A9 infection. The left and right Y-axis represent the mean percentage of virus yield and cytotoxicity inhibition, respectively. (**C**) 2’-FL blocks the infection without destroying the virus structure. CV-A9 was pre-incubated with DMEM or 2’-FL (10 mg/mL) for 2 h before plaque assay. Both treatment groups exhibited the same viral titres. MOI: multiplicity of infection; qRT-PCR: quantitative real-time polymerase chain reaction; EC_50_: concentration for 50% of maximal effect; CC_50_: cytotoxicity concentration 50%; mRNA: messenger RNA. The experiments were repeated at least twice. Data represent the mean of the three replicate results, and error bars represent ±1 SD; ns: no significant difference; Student’s *t*-test compared with DMEM.

**Figure 3 ijms-23-13727-f003:**
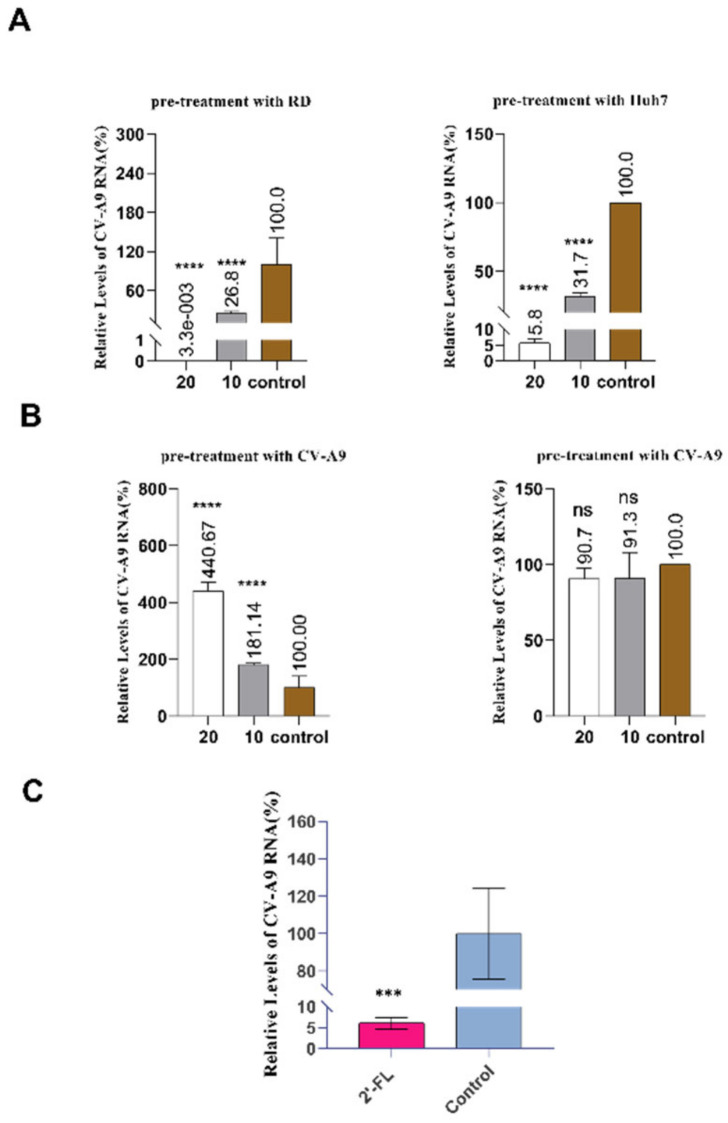
2’-FL acts on cells to block infection. (**A**) Cells were incubated with 2′-FL at a final concentration of 20 mg/ml and 10 mg/ml for 4 h at 37 ℃, followed by CV-A9 infection (MOI = 0.001) in the presence of 2’-FL at the same concentration. At 48 h.p.i., the intracellular viral load was quantified using qRT-PCR and normalised by GAPDH level. (**B**) CV-A9 were incubated with 2’-FL at a final concentration of 20 mg/ml and 10 mg/ml for 4 h at 4 ℃ and added into RD cells at MOI = 0.001. At 48 h.p.i., the intracellular viral load was quantified using qRT-PCR and normalised by GAPDH level. (**C**) 2’-FL (20 mg/ml) was used to pre-treat RD cells for 24 h. Then the supernatant was removed and replaced with CV-A9 virus solution (MOI = 0.001) for 24 h. Cells without drug treatment but only with virus treatment were the control group. The intracellular viral load was quantified using qRT-PCR and normalised by GAPDH level. **** *p* < 0.0001; *** *p* < 0.001; ns: no significant difference; Student’s *t*-test compared with control. Data represent the mean of the three replicate results. Error bars represent ±1 SD, and the experiments were repeated at least twice.

**Figure 4 ijms-23-13727-f004:**
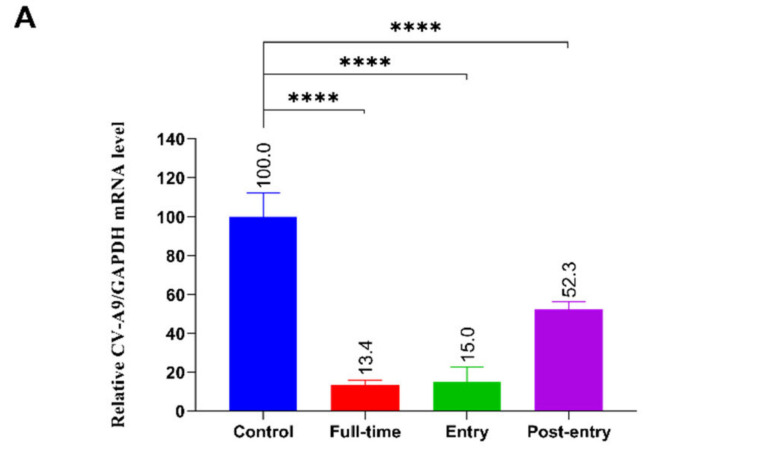
2’-FL inhibits CV-A9 infection by blocking viral cell entry. (**A**) Time-of-addition experiment of 2’-FL (10 mg/mL). The detailed steps for the time-of-addition experiment were described in the method section, the viral and GAPDH mRNA levels in the infected cell were quantified using qRT-PCR at 24 h.p.i. The viral load in the supernatant was measured by plaque assay. Time course curve (**B**) and single-step growth curve (**C**) of CV-A9 production (top) in RD cells with or without a single dose of 2’-FL (10 mg/mL) or LF (0.156 mg/mL) treatment after 2 h.p.i. Supernatant and cells were harvested at 4, 6, 10, 12, 24, 36, and 48 h.p.i. for qRT-PCR. Plaque assay corresponding to the CV-A9 single-step growth curve (bottom). (**D**,**E**) Illustrates the time periods when 2’-FL was present in the cell culture. The arrows indicate the time period 2’-FL (10 mg/mL) exists in the cell culture (left). RD cells were infected with CV-A9 at MOI = 5, with cells following the corresponding drug treatment. At 12 h.p.i., the viral yield in cells was quantified using qRT-PCR and normalised by GAPDH level (right). Determining the effectiveness of 2’-FL on (**F**) viral attachment and (**G**) viral internalisation. Virus infection and drug treatment were performed as mentioned above. Briefly, RD was infected with CV-A9 at 4 ℃ for 2 h and then at 37 ℃ for 1 h, allowing temperature-sensitive entry steps to proceed. p.i.: post-infection. The shown results are representative of one experiment out of at least two experiments. Data represent the mean of the three replicate results, error bars represent ±1 SD; ns: no significant difference; **** *p* < 0.0001 by Student’s *t*-test compared with control.

**Figure 5 ijms-23-13727-f005:**
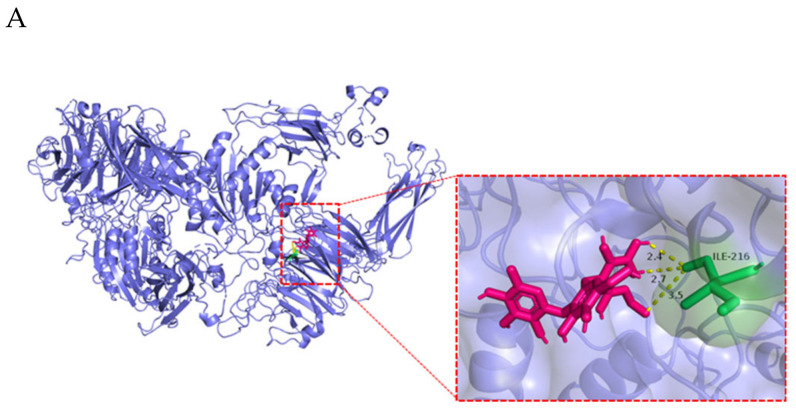
Docking analysis visualisation of 2’-FL binding with (**A**) ITGB6 (PDB: 4UM8), (**B**) FCGRT (PDB: 1EXU), and (**C**) β_2_M (PDB: 1A1M). The yellow dots show H-bonds. The ITGB6-2’-FL complex structure highlighting the ITGB6 (blue), 2’-FL (magenta) and residue I216 (green); the FCGRT-2’-FL complex structure highlighting the FCGRT (blue), 2’-FL (magenta) and residue W51 (green); The β_2_M-2’-FL complex structure highlighting the β_2_M (blue), 2’-FL (magenta) and residue Q155, D4 and R62 (green).

**Figure 6 ijms-23-13727-f006:**
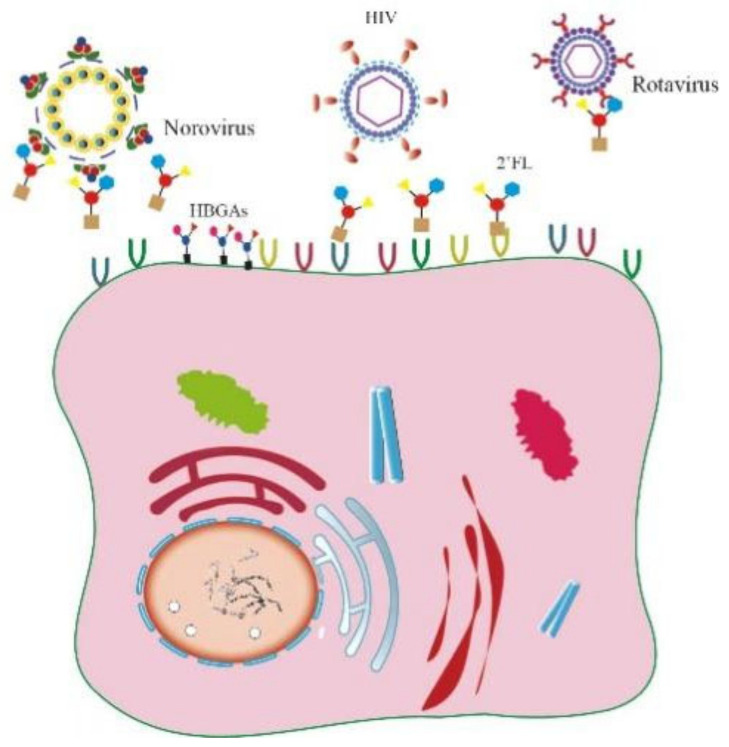
Schematic overview on the antiviral role of 2’-FL. Similar to cell surface glycan (such as HBGA), 2’-FL can act as a soluble decoy receptor to bind to the virus to block infection; 2’-FL can also block viral attachment by binding to cell surface receptors.

**Table 1 ijms-23-13727-t001:** Molecular docking analysis of several compounds against 2’-FL.

Ligand	Protein	Binding Energy (ΔG) kcal/mol	Ligand Efficiency
2’-FL (C_18_H_32_O_15_)	FCGRT	−2.34	−0.07
	β_2_M	−3.19	−0.1
	ITGB6 (α_v_β_6_)	−2.14	−0.06

**Table 2 ijms-23-13727-t002:** Oligonucleotides used in the study.

Oligonucleotide Name	Sequence
CV-A9 -F	TCATGACACCAGCTGATAAG
CV-A9-R	TGCTCATCTGCTCTGAAGTATC
GAPDH-F	AGCCTCAAGATCATCAGCAATG
GAPDH-R	ATGGACTGTGGTCATGAGTCCTT

## Data Availability

All dates generated in this study are included in the article/Appendix A.

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
