# Peer review of "2’-Fucosyllactose Inhibits Coxsackievirus Class A Type 9 Infection by Blocking Virus Attachment and Internalisation"

_ijms, 2022, doi:10.3390/ijms232213727_

Round 1
Reviewer 1 Report
The authors screened 16 major components of the human milk against coxsackievirus class A type 9 isolate. Using various in vitro assays, Fucosyllactose (2'-FL), one of the components of human milk, was found to inhibit coxsackievirus replication at various stages of its life cycle. Finally, using molecular docking studies, it was shown that 2'-FL interacts with the attachment receptor αvβ6 and the internalization receptor FCGRT, β2M. While the study is interesting, data representation and explanation need more improvement. The methods section also needs improvement.
1. Line 9 - Human milk contains various biologically active pathogens. Is it pathogens, or it contains biologically active components against pathogens?
2. Viral inhibition levels were shown in percentages in several places, but it is unclear. For example, in lines 16-17, compared with the control group, 2'-FL blocked virus entry (infection%: 100% vs 15%). It will be easy to understand if we interpret them as 85%.
3. Positive controls were mentioned in several places. Again it is not very clear, for example, in Figures 1 and 4B. Positive control means a known compound used as a positive control to show viral inhibition. However, here it is not a positive control. Is it cells treated with virus and vehicle?
4. Lines 122-126- Pre-treating the virus with 2'-FL, there was no effect.
A) Please describe how the experiment was performed in the methods section.
B) The pretreated virus also contains 2'-FL. Why did it not show any effect?
5. Lines 159-162- The authors claim no effect of 2'-FL on viral genome replication.
A) Please describe how the time course and single-step growth curve experiment were performed in the methods section.
B) Viruses from infected cells are released and subsequently attach and enter new target cells for virus replication. If 2'-FL inhibits viral entry, the virus cannot infect new target cells. Hence, the percentage of infected cells should be reduced (Fig 4B). It is unclear how this experiment was performed.
Author Response
Reviewer #1 (Comments and Suggestions for Authors):
The authors screened 16 major components of the human milk against coxsackievirus class A type 9 isolate. Using various in vitro assays, Fucosyllactose (2'-FL), one of the components of human milk, was found to inhibit coxsackievirus replication at various stages of its life cycle. Finally, using molecular docking studies, it was shown that 2'-FL interacts with the attachment receptor αvβ6 and the internalization receptor FCGRT, β2M. While the study is interesting, data representation and explanation need more improvement. The methods section also needs improvement.
Response:
Thanks for your approbation of this manuscript, and we appreciate your constructive suggestions. We have addressed the issues you pointed out to improve the quality of our manuscript.
- Line 9 - Human milk contains various biologically active pathogens. Is it pathogens, or it contains biologically active components against pathogens?
Response:
Thank you for your kind reminder, we have corrected the original sentence and the revised sentence is “Human milk contains various biologically active components against pathogens.” (line 9)
- Viral inhibition levels were shown in percentages in several places, but it is unclear. For example,in lines 16-17, compared with the control group, 2'-FL blocked virus entry (infection%: 100% vs15%). It will be easy to understand if we interpret them as 85%.
Response:
Thank you for your kind reminder, we have made changes as required, making it clearer and easier to understand. (line17-18, 21, 127, 131, 139, 204)
- Positive controls were mentioned in several places. Again it is not very clear, for example, inFigures 1 and 4B. Positive control means a known compound used as a positive control to show viralinhibition. However, here it is not a positive control. Is it cells treated with virus and vehicle?
Response:
Thank you for your kind reminder, we have modified the manuscript and figures according to your reminder by changing “positive control” to “control” to indicate that the cells are not treated with drugs but only with virus.
- Lines 122-126- Pre-treating the virus with 2'-FL, there was no effect.
- A) Please describe how the experiment was performed in the methods section.
Response:
Thank you for your kind reminder, we have added a specific description of the experiment in the methods section. Cells were co-incubated with 2'-FL for 4h at final concentrations of 20 mg/ml, 10 mg/mL respectively at 37 °C, followed by infection with CV-A9 (MOI=0.001) in the presence of the same dosages of 2'-FL as before. At 2 h.p.i., the virus–drugs mixture was removed, and a fresh culture medium containing the same drug dosages as before was added to each well. At 48 h.p.i., all cells were harvested for RNA extraction. After pre-diluting the virus solution to MOI=0.002, the virus solution was incubated with 2'-FL (volume ratio 1:1) for 4h at 4 °C to ensure the final drug concentration of 20 mg/ml and 10 mg/ml respectively, and the final MOI is 0.001. The virus-drugs mixture was then added to the cells. At 2 h.p.i., the virus–drugs mixture was removed, and a fresh culture medium containing the same drug dosages as before was added to each well. At 48 h.p.i., all cells were harvested for RNA extraction. Cells without drug treatment but only with virus treatment was the control group. (line343-354)
- B) The pretreated virus also contains 2'-FL. Why did it not show any effect?
Response:
Thanks for your question. Our experiments confirm that 2'-FL acts mainly at the viral entry stage, which means that the drug acts at the early infection stage of the virus (0-2h). Moreover, it mainly targets receptors on cells, so pre-incubation of drug with cells allows 2'-FL to interact with cellular receptors earlier to competitively inhibit viral binding. However, in the drug-virus preincubation group, we hypothesized that there is a weak interaction between 2'-FL and virus, which is not targeting the viral active site and therefore cannot block viral binding to cell receptors. However, this weak interaction between virus and 2'-FL promotes viral arrival at the cell surface, thus increasing viral binding to the cell. The similar phenomenon was also reported for G10P rotavirus, where treatment with the same dose of mixed HMOs reduced G3P rotavirus infection but enhanced G10P rotavirus (https://doi.org/10.1038/s41467-018-07476-4). We have provided additional explanations in the discussion section according to your reminder. (line267-274)
- Lines 159-162- The authors claim no effect of 2'-FL on viral genome replication.
- A) Please describe how the time course and single-step growth curve experiment were performedin the methods section.
Response:
Thank you for your kind reminder, we have added a specific description of the experiment in the methods section. Cells were co-incubated with CV-A9 (MOI=0.001). At 2 h.p.i., the supernatant was removed, and cells were washed twice with PBS to remove unbound viruses. Then a fresh culture medium containing 2'-FL (10 mg/mL) or LF (0.156 mg/mL) was added to each well. Cells were harvested at 4, 6, 10, 12, 24, 36, and 48 h.p.i. for qRT-PCR and time course curves assay. The supernatant harvested at 4, 6, 10, 12, 24, 36, and 48 h.p.i. contains live progeny virus, enabling analysis of single-step growth curves by plaque assay. (line392-398)
- B) Viruses from infected cells are released and subsequently attach and enter new target cells forvirus replication. If 2'-FL inhibits viral entry, the virus cannot infect new target cells. Hence, thepercentage of infected cells should be reduced (Fig 4B). It is unclear how this experiment was
Response:
Thank you for your suggestion, we have added a specific description of the experiment in the methods section. And in our experiments, cells and supernatants were collected for viral mRNA expression levels and live progeny virus load determination, respectively. At 2h.p.i. the virus completes the cell entry process, followed by drug treatment. The viral load remained stable at 2-4 h.p.i. and increased significantly at 4-12 h.p.i., suggesting the initiation of viral replication, as shown in the intracellular viral changes curve (Fig 4B, time course curve. From the live virus growth curve in the supernatant (Fig 4C, single-step growth curve), we can see that the increase of live virus load occurred after 6 h.p.i., indicating viral release. The 2-12 h.p.i. in the time course and single-step growth curves represent one round of the virus life cycle, in which the virus completes the entire process from replication, assembly to release. It is evident from the time course and single-step growth curves that 2'-FL did not inhibit viral replication in 2-12 h.p.i. In addition, as seen in Fig 4C, the amount of live virus in the supernatant can increase 100-fold with each completed round of release. Therefore, 2'-FL is not dominant in subsequent competitive inhibition to block virus-cell binding, and relative quantification is used in Fig 4B, where weak inhibition may be overlooked. The vertical coordinate in Fig 4C is p.f.u.ml-1(log10), which was measured by plaque assay, and it can be seen that at 48 h. p.i. 2'-FL can only weakly reduce the live virus load.

Reviewer 2 Report
In this manuscript titled “2'-Fucosyllactose inhibits coxsackievirus class A type 9 infection by blocking virus attachment and internalization”, the authors depict the inhibitory effect of 2’-Fucosyllactose (2’-FL), a human breast milk component, on coxsackievirus class A type 9 isolate. The authors demonstrate that 2’-FL blocks viral entry, attachment and internalization. Following suggestions could be incorporated to improve the quality of the manuscript.
1) The resolution quality for most of the figures is not good, the authors should replace these images with exporting images at a better resolution.
2) How many replicates are used for the data represented in the manuscript? It should be mentioned clearly in the text and figure legends. Information about statistical analysis should be briefly mentioned in the figure legends.
3) What is the negative control or the no treatment control? What vehicle were the drugs dissolved in? Do the authors have vehicle only controls?
4) The drug (2’-Fucosyllactose) concentration used for the experiments are 10mg/ml (20mM) or 20mg/ml (40mM) are quite high, are these physiologically feasible?
5) The CC50 plot is hard to interpret, can the authors consider replacing % cytotoxicity with % cell viability? In my opinion, that will make it easier to interpret the results.
6) Line 67 starting with “The final..” should be reframed.
7) It is not clear from Figure 1, which assay was performed to quantify viral RNA. It should be made clear in the text and in the figure legends.
8) Figure 1, the plot legends mention “Relative”, what is the reference here?
9) In Figure 1, do the authors also have information about the cytotoxicity for different drugs tested?
10) In Figure 2A, the plot legend seems to be mislabeled as Inhibitor Rate% where it should be Inhibition rate%.
11) In Figure 2A, there is no negative + vehicle control.
12) In Figure 2, no information about replicates and no statistical analysis performed for 2B or C.
13) In Figure 3A, there is a spelling error in one of the plot headings.
14) In Figure 3A and B, no significance testing is performed.15) The authors depict that 2’FL affects the attachment and entry, can the author use a VSV-G pseudotyped virus to show that?
Author Response
Responses to
Reviewer #2 (Comments and Suggestions for Authors):
In this manuscript titled “2'-Fucosyllactose inhibits coxsackievirus class A type 9 infection by blocking virus attachment and internalization”, the authors depict the inhibitory effect of 2’-Fucosyllactose (2’-FL), a human breast milk component, on coxsackievirus class A type 9 isolate. The authors demonstrate that 2’-FL blocks viral entry, attachment and internalization. Following suggestions could be incorporated to improve the quality of the manuscript.
Response:
Thanks for your approbation of this manuscript, and we appreciate your constructive suggestions. We have addressed the issues you pointed out to improve the quality of our manuscript.
- The resolution quality for most of the figures is not good, the authors should replace these images with exporting images at a better resolution.
Response:
Thank you for your kind reminder, and we have already replaced these images with a better resolution.
- How many replicates are used for the data represented in the manuscript? It should be mentioned clearly in the text and figure legends. Information about statistical analysis should be briefly mentioned in the figure legends.
Response:
Thank you for your kind reminder. The shown results are representative of one experiment out of at least two experiments, and data represent the mean of the three replicate results. And we have indicated the replicates of the data as well as the statistical analysis information in the text and figure legends. (line81-82, line109-111, line153-154, lin183-186)
- What is the negative controlor the no treatment control? What vehicle were the drugs dissolved in? Do the authors have vehicle only controls?
Response:
Thanks for your question, we have adjusted the “positive control” in the text, figures and figure legends to “control” to indicate that the cells are not treated with drugs but only with virus. “Negative control” and “no treatment control” indicates the cells was treated with only the vehicle for drug dissolution. We have defined the specific meaning of “control” in the text to prevent ambiguity (line150-151). The compounds in our experiments were dissolved in DMEM (we have clarified in the text based on your reminder) and showed good solubility in the concentration range used in our experiments. We set up vehicle only controls in the experiment to observe whether the cells are in good condition or not. Since all the vehicles in the experiment were DMEM, and the dilution of the viral solution was also performed by DMEM, the antiviral effect of vehicles could be excluded.
- The drug (2’-Fucosyllactose) concentration used for the experiments are 10mg/ml (20mM) or 20mg/ml (40mM) are quite high, are these physiologically feasible?
Response:
Thanks for your question, concentrations of 10 mg/ml (20 mM) or 20 mg/ml (40 mM) of 2'-FL were not physiologically feasible. One study analyzed 2’-Fucosyllactose concentrations in breast milk using LC-MS/MS and showed that 2’-Fucosyllactose concentrations were generally in the range of 0.4-2.6 g/L (https://doi.org/10.1007/s10068-022-01154-4). However, Our data show that 2'-FL without cytotoxicity at 10 mg/ml and 20 mg/ml (Fig 2B), and the European Union (EU) considers two HMOs, 2’-FL and LNnT, novel foods (Commission Implemented Regulation (EU) 2017/2470). Besides, our data show that reduced concentrations of 2'-FL (5 mg/ml) still exhibit 99% inhibition against CV-A9 on RD cells (Fig 2A). Therefore 2'-FL can be added in moderate amounts as a food or pharmaceutical supplement.
- The CC50 plot is hard to interpret, can the authors consider replacing % cytotoxicitywith % cell viability? In my opinion, that will make it easier to interpret the results.
Response:
Thank you for your kind reminder, we used % cell viability instead of % cytotoxicity to make the results clearer according to your suggestion.
- Line 67 starting with “The final..” should be reframed.
Response:
Thanks for your suggestion, we have made corrections in the article. The revised sentence is “The final concentrations of all components used for primary screening have been indicated in the methods section.” (line 68-69)
- It is not clear from Figure 1, which assay was performed to quantify viral RNA. It should be made clear in the text and in the figure legends.
Response:
Thanks for your kind reminder, and we have described the analytical methods for quantifying viral RNA in the figure legend (line78-81) and text (line 324-328)
- Figure 1, the plot legends mention “Relative”, what is the reference here?
Response:
Thanks for your question. The relative viral mRNA expression and inhibition rate were calculated based on the Ct value, denoted by B and I, respectively. The formula is as follows: A = expression level of CV-A9 mRNA / GAPDH mRNA = 2^(CtGAPDH-CtCV-A9); Bingredient (%) = (Aingredient / Acontrol) × 100%; Iingredient (%) = 1 - (Bingredient - Bcontrol) × 100%. Therefore, the plot legends “relative” indicates the percentage of CV-A9 mRNA level in drug-treatment group relative to control.
- In Figure 1, do the authors also have information about the cytotoxicity for different drugs tested?
Response:
Thanks for your suggestion, we have re-supplemented the cytotoxicity tests for all drugs at the primary screening concentrations and the results are shown as cell plots in the supplementary figure 3. All components showed no significant cytotoxicity at the primary screening concentrations.
- In Figure 2A, theplot legend seems to be mislabeled as Inhibitor Rate% where it should be Inhibition rate%.
Response:
Thanks for your suggestion, we have modified the figure according to your suggestion.
- In Figure 2A, there is no negative + vehicle control.
Response:
Thanks for your kind reminder, we have modified the figure according to your suggestion. “Negative + vehicle control” in Figure 2A is indicated by “DMEM”.
- In Figure 2, no information about replicates and no statistical analysisperformed for 2B or C.
Response:
Thanks for your kind reminder, we have performed statistical analysis and noted the replicates in the figure legend. The experiments were repeated at least twice, data represent the mean of the three replicate results, and error bars represent ±1 SD; ns, no significant difference, Student’s t-test compared to DMEM.
- In Figure 3A, there is a spelling error in one of the plot headings.
Response:
Thanks for your kind reminder, and we have corrected the spelling error to “pre-treatment with RD”.
- In Figure 3A and B, no significance testing is performed.
Response:
Thanks for your kind reminder, we have performed significance testing on the figures and performed statistical analysis in the figure legend. . ****p<0.0001; ***p<0.001; ns, no significant difference, unpaired Student’s t-test compared to control. Data represent the mean of the three replicate results. Error bars represent ±1 SD, and the experiments were repeated at least twice.
- The authors depict that 2’FL affects the attachment and entry, can the author use a VSV-G pseudotyped virus to show that?
Response:
Thanks for your kind suggestion. As we all know, viral fusion proteins are essential for enveloped virus infection. The vesicular stomatitis virus G (VSV-G) protein is a typical type III viral fusion protein. Simplified assays using VSV-G pseudovirus have been applied to detect cell entry processes of various enveloped viruses, including influenza, severe acute respiratory syndrome coronavirus (SARS-CoV), human immunodeficiency virus (HIV) etc. However, CV-A9 is a non-enveloped virus and little is known about the packaging signal of non-enveloped viruses, which has made it difficult to perform VSV-G pseudovirus particle packaging to date. In our experiments, we confirmed that 2'-FL can inhibit virus attachment and entry by attachment and internalization experiments, and the method has been reported in other studies (DOI: https://doi.org/10.1128/mBio.02342-21) (DOI: 10.1128/jvi.02042-21), which indicates the validity of the method.

Round 2
Reviewer 2 Report
The authors did a great job in addressing all the concerns. A small suggestion about the picture quality, the text quality in the figures is still sub-optimal and hard to read, it will be great if that can be improved.